# Soluble and multivalent Jag1 DNA origami nanopatterns activate Notch without pulling force

Ioanna Smyrlaki[1], Ferenc Fördős [1], Iris Rocamonde-Lago[1], Yang Wang [1], Boxuan Shen [1,2], Antonio Lentini [1], Vincent C. Luca [3], Björn Reinius [1], Ana I. Teixeira [1] & Björn Högberg [1] ✉

The Notch signaling pathway has fundamental roles in embryonic development and in the nervous system. The current model of receptor activation involves initiation via a force-induced conformational change. Here, we define conditions that reveal pulling force-independent Notch activation using soluble multivalent constructs. We treat neuroepithelial stem-like cells with molecularly precise ligand nanopatterns displayed from solution using DNA origami. Notch signaling follows with clusters of Jag1, and with chimeric structures where most Jag1 proteins are replaced by other binders not targeting Notch. Our data rule out several confounding factors and suggest a model where Jag1 activates Notch upon prolonged binding without appearing to need a pulling force. These findings reveal a distinct mode of activation of Notch and lay the foundation for the development of soluble agonists.

Notch signaling is an evolutionary conserved cell-to-cell communication system present in most animals with fundamental roles in cell fate decisions, tissue patterning, angiogenesis, and neurogenesis[1–3]. In mammals, the extracellular part of the Notch (NOTCH1-4) receptors (NECD) is composed of 29-36 epidermal growth factor (EGF) repeats (Fig. 1a)[4,5]. The exact function of these repeats is not well understood and whether the repeat-region is rod-like (Fig. 1a, left) or if it could possess a folded tertiary structure (Fig. 1a, right) is unclear. Activation of the receptor relies on three proteolytic steps performed on three cleavage sites in the Notch receptor (S1-S3) (Fig. 1b): one cleaved during maturation (S1), a second (S2) cut by ADAM metalloprotease following binding and third, a transmembrane γ-secretase (S3) mediated cut resulting in release of the Notch intracellular domain (NICD)[1,2,6]. The NICD then translocates to the nucleus to form a transcriptional activator complex[3].

Following the EGF repeats in the NECD, are three cysteine-rich Lin12-Notch repeats (LNR) and a hydrophobic heterodimerization domain at the pre-cleaved S1 that together form what is known as the negative regulatory region (NRR)[7]. The NRR is close to the cell membrane and is shown to form a loop that is hypothesized to protect the S2 site from unregulated ADAM cleavage[8,9]. The Notch ligands (Serrate: JAG1-2 and Delta: DDL1,3,4) are also membrane proteins that have been studied extensively[4,10,11]. In the canonical model of pathway activation, these are typically endocytosed by the ligand-expressing cell after binding to Notch[12,13]. The dominant theories for this endocytosis are (i) a receptor recycling model where each receptor (and ligand) is used only once, and (ii) a model in which a pulling force would be required to uncover the S2 region of the receptor.

That a force of about 4-12pN can activate Notch receptors has been shown experimentally using tension gauge tethers, magnetic tweezers, and force clamp spectroscopy[11,14–16]. The former indicating a catch bond behavior for both JAG1 and DLL4[11]. It is also clear, from the structure of the receptor, that excessive pulling[16] or disruption of the heterodimerization domain by chelators[17] will shed the ECD and expose the receptor for activation.

However, the notion that a pulling-force is strictly necessary is complicated by other experiments that indicate long-range activation via secreted ligands[18,19] and by in vitro activation using beads or Fc-clustered ligands[20,21] as well as being complicated by recent results on synthetic Notch constructs[22–24]. There has also not been successful

[1]Department of Medical Biochemistry and Biophysics, Karolinska Institutet, Stockholm, Sweden. [2]Department of Bioproducts and Biosystems, School of Chemical Engineering, Aalto University, Alto, Finland. [3]Department of Immunology, Moffitt Cancer Center, Tampa, FL, USA. ✉e-mail: bjorn.hogberg@ki.se

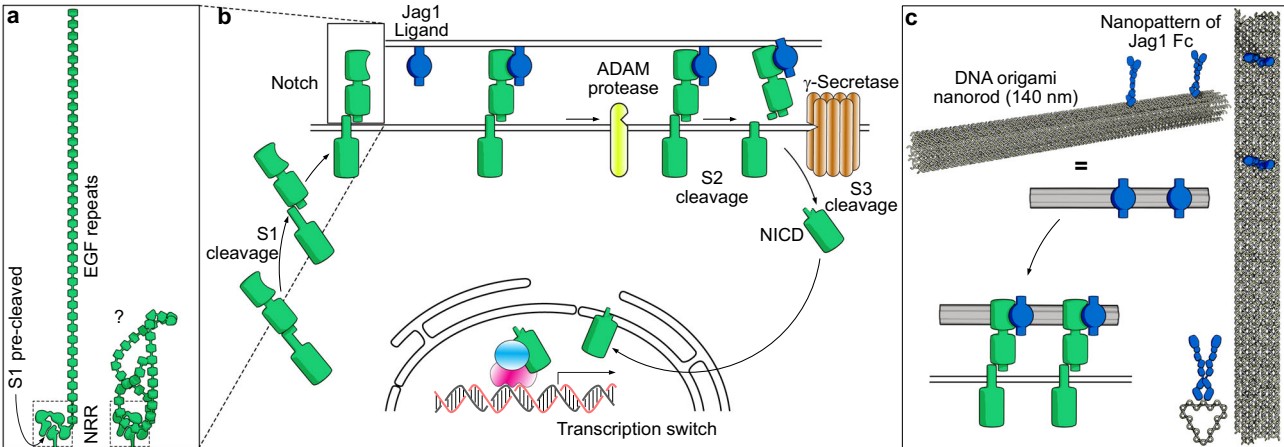

**Fig. 1 | Schematic view of the Notch pathway and experimental principle. a** The extracellular domain of the Notch receptor comprises a negatively regulatory region (NRR, dashed box) containing a pre-cleaved S1 site held together non-covalently and 29-36 (depending on Notch subtype) EGF repeats. Whether these are stretched out (left) or form tertiary structures (right), is unclear. **b** The canonical Notch pathway. Notch heterodimers are formed via Furin cleavage of the S1 in the Golgi, followed by membrane insertion. Binding to ligands induces cleavage of

S2 by ADAM proteases, which enables cleavage of S3 via γ-Secretase to release the Notch intracellular domain (NICD). The NICD translocates to the nucleus forming a transcription activator complex with CSL and Mastermind. **c** DNA origami nano-patterns of Jag1 were designed where Jag1-Fc molecules covalently conjugated to DNA oligonucleotides could be attached to the nanostructures to form many types of precisely controlled multivalent Jag1 binder patterns.

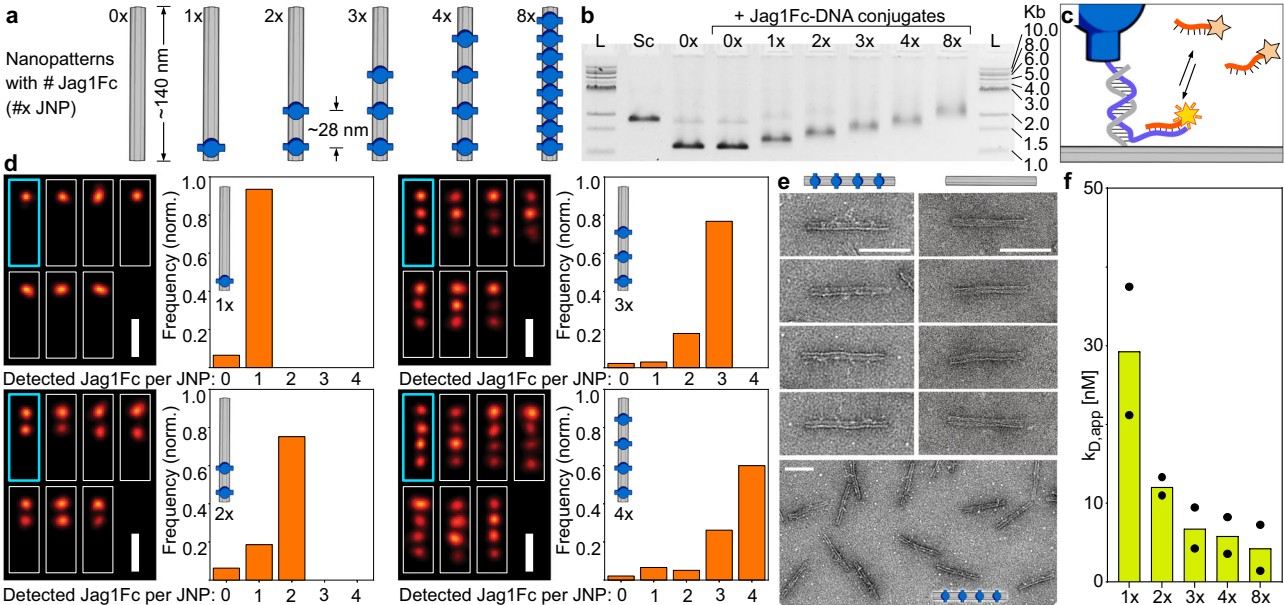

**Fig. 2 | Characterization of Jag1Fc DNA nano-patterns. a** A rod like DNA origami was used to create 1-8x Jag1Fc nanopatterns (JNPs). **b** Gel shift assay. 1Kb ladder (L), origami scaffold alone (Sc), empty JNP(0x), a repeat of(0x) but with added Jag1Fc-DNA conjugates (control for non-specific binding), structures loaded with Jag1Fc patterns: 1-8x JNP (2% agarose, EthBr stained). Agarose gel repeated independently for *n* = 20 **c** Schematic representation of the DNA origami used for the DNA PAINT experiments. Here, the Jag1-DNA conjugates also contain an extension for DNA-PAINT docking sites. **d** DNA PAINT on Jag1Fcs on DNA origami Average (cyan, thick)

and individual cropped DNA PAINT superresolution images (white, thin) of JNPs (scale bars = 50 nm). Bar graphs show the Jag1 site occupancy distributions of the different nanorods. **e** Zoom-ins of negative stain TEM of (unpurified) 4x JNPs and empty DNA origami rods (0x JNP) and zoom-out of 4x JNPs (bottom). Scale bars are 100 nm. Micrographs were repeated independently for *n* = 5 **f** Receptor immobi-lized on SPR chip surface and increasing concentrations of 1-8x JNP as analyte. The bar graph shows the mean apparent kD of different Jag1Fc nano-patterns, dots represent two individual repeats.

targeting of the pulling mechanism itself in vivo or in therapeutics. While patterns of Notch ligands have been investigated on the microscale and on surfaces[25,26], precise patterns displayed from the solution have not been tested. To re-investigate the molecular mechanism and to look at receptor activation from a solution phase in detail, we designed precise DNA origami nanostructures[27] (Fig. 1c) that display variations of patterns of the Notch ligand Jag1 and used these to stimulate endogenous Notch in human neuroepithelial-like stem cells.

## Results

### Achieving controlled Jag1 nanopatterns

We used a rod like DNA origami structure[28] (Fig. 1c, Methods) to spatially position zero (baseline-control), one, two, three, four or eight, dimeric Jag1Fc fusion proteins (Fig. 2a) and study the ability of nanoscale ligand clusters to activate the Notch signaling pathway. To assemble the proteins with the DNA origami, we hybridized the folded DNA nanos-tructures, containing single-strand protruding oligos at desired posi-tions, to site-specifically labeled Jag1Fc-DNA conjugates added in excess,

and then purified the structures using size exclusion (Methods). Using a gel retardation assay, we confirmed the successful production of Jag1 nanopatterns (JNP), where an increasing shift was observed as more proteins were hybridized onto the DNA origami nanorod (Fig. 2b).

We used DNA-PAINT (DNA points accumulation for imaging in nanoscale topography)[29] to image and validate the functionalization states of the different JNPs. For this experiment only, the DNA nanostructures were prepared with additional biotin handles for immobilization in the imaging chamber and Jag1 conjugates carrying DNA-PAINT docking sites for direct detection of proteins on the structures with DNA PAINT imaging (Fig. 2c, Supplementary Fig. 1, Methods). For all structures the fraction with the highest detected frequency was the one with the designed number of proteins with the mean functionalization state being also close to the designed (Fig. 2d). Furthermore, the majority of the probes were present as clear monomers (72.6%), with negligible curvature and site-to-site distances were closely resembling the designed distances at approx. 28 nm separations between Jag1Fcs (Supplementary Fig. 1).

To further confirm the correct geometry of the structures we imaged the empty- and the 4x JNPs- structures with negative stain transmission electron microscopy (TEM) (Fig. 2e). The Jag1Fc are barely detectable with TEM due to the molecule being thin with only the small Fc region being globular, as well as the molecule being on the low molecular weight end for negative stain TEM imaging. Overall, the gel assays, DNA-PAINT analysis, and TEM imaging all confirm that we could produce well-controlled monodisperse nanostructures with precise incorporation of Jag1Fc.

To further examine how the ligand multivalency affected binding kinetics in vitro, we used surface plasmon resonance (SPR). We immobilized a biotinylated Notch1 (EGF8-12) protein on a streptavidin modified chip surface and increasing concentrations of JNPs were injected onto the chip. By performing multi-cycle kinetics analysis, we found that the apparent $k_D$ of the JNPs decreases when the number of the proteins per nanopattern increases (Fig. 2f). Furthermore, the SPR measurements revealed faster association and slower dissociation when more proteins were loaded on the JNPs (Supplementary Fig. 2).

## Jag1 multivalency drive increasing Notch activation

We used long-term self-renewing neuroepithelial-like stem cells (lt-NES cells) which are a model of early neuronal progenitors and stimulated these cells from solution with versions of the JNPs. (Fig. 3a). Notch activity has been reported in these iPSc-derived cells (Methods)[30] where high levels of Notch related genes like *HES1* and *HES5* could be observed while inhibition of the Notch pathway in this system led to neuronal differentiation[30,31]. The model system is motivated by the resemblance of progenitors of the early neural tube, where Notch is believed to be highly relevant for developmental patterning[32]. Importantly, the Notch genes in these cells are not genetically engineered to give signal amplification, and thus provide relevant indications of how an endogenous Notch pathway react to multivalency. We confirmed the presence of Notch1-3 using immunostaining (Fig. 3b) and RNA sequencing (Supplementary Fig. 3). We screened and picked a cell seeding density (19k cells/cm²) where our experiment was not confounded by a large portion of Notch activation mediated by cell-cell interactions (Supplementary Fig. 4a). In most assays, we used qPCR to measure *HES1* expression levels compared to the housekeeping gene *GAPDH*. A time course experiment using 3x JNPs between 2–7 hrs showed a high activation level at 3 h after initiation of JNPs stimulation (Fig. 3c). The 3 hr time-point was used in the following experiments. Our data follow a known dynamic expression profile of Notch target genes. This is an important effect during developmental processes and in the maintenance of neuronal progenitors, whereas stable expression inhibits both processes[33–35].

Importantly, to compare the effect of different JNPs on Notch activation, we normalized their concentrations to the proteins, and not to the molarity of DNA origami itself, assuming all nanostructures have all Jag1Fc attached (Note that this method of normalizing likely underestimate the amount of total ligands, particularly in the JNPs with a high number of Jag1Fc, due to incomplete labeling, see Fig. 2b, d). First, we treated the iPS cells with 3.3 nM of total protein on 1-4x JNPs and observed a significant upregulation when more than one protein was placed per nanopattern, with increasing number of Jag proteins per structure resulting in increasing activation (Fig. 3d).

Looking at dosage dependence, we observed that for every case, cells reached a plateau of stimulation after 1.7 nM Jag1Fc (Fig. 3e). Again this corroborates that the multivalency appears to be the most important factor. We also validated that the upregulation of *HES1* was an effect of Notch receptor activation, by seeing no effect of JNP stimulation after adding inhibitors for ADAMs, and γ-Secretase, respectively (Supplementary Fig. 4b).

Direct verification of activation of endogenous Notch receptors have been challenging due to a lack of tools available in other receptor systems, for example, such as those for tyrosine kinase-activated receptors. Here we applied a proximity ligation assay (PLA), that amplifies the signal from an antibody against a region that is exposed only after γ-secretase cleavage occurs (Methods). We observed significantly more PLA signal per cell when stimulated with the 8x JNP (Fig. 3f, g and Supplementary Fig. 4c), verifying again that the detected downstream effects were indeed due to Notch being activated.

To further analyze downstream signaling, we stimulated iPS cells with 0x (empty JNPs), 1x, and 8x JNPs, in three different biological repeats and analyzed the samples with RNA sequencing (RNA-seq). Clusters revealed using hierarchical complete-linkage clustering, show distinct profiles for the three cases (Fig. 3h). Importantly, RNA-seq revealed that many Notch-related genes were upregulated after stimulation with JNPs (Fig. 3i, j). Interestingly, we observed that genes that have previously been shown in other cellular systems to be dependent on dimerization of NICD at the Notch transcription complex (NTC) e.g. *HES1* and *HES5*, became upregulated by larger clusters of Jag1, while *HEY1*, which likely depends on monomeric NICD transcription complex, reached its maximum upregulation already with 1x JNP[36,37].

## Multivalency effect persists irrespective of potential force sources

Considering that a repulsion force could be generated between our Jag1 nanopatterns and the cell membrane due to their mutual negative charge (Fig. 4a), we shielded the charge of the DNA origami JNPs by coating it with an oligolysine (K10) solution to reduce the negative surface charge of the structures and thus to minimize any potential repulsive forces between these and the lipid membrane (Methods, Supplementary Figs. 5a and 6). We observed that shielding the negative charge of the DNA nanostructures did not reduce the relative *HES1* gene levels, indicating that electrostatic repulsive forces did not mediate Notch activation in our system (Fig. 4a).

There could further be a force generated by cells moving across structures that are non-specifically bound to the cell culture surface (Fig. 4b). To investigate this, we used an assay to probe trace concentrations on surfaces of the DNA origami scaffold using qPCR (Methods). By this quantification, we could observe only traces of DNA origami non-specifically bound, and notably the measurements were below the linear range of the assay (Fig. 4b). The K10 coated origami showed an even lower surface attachment. Combined with the results above, this strongly implies that the multivalency effect we observed in our Notch stimulation, cannot be explained by random surface attachment of JNPs.

It has been hypothesized that one way of generating forces in the Notch signaling system could be from forces generated by invaginations of the cell membrane following endocytosis of the receptors (Fig. 4c). To investigate this, we stimulated IPS cells as above with

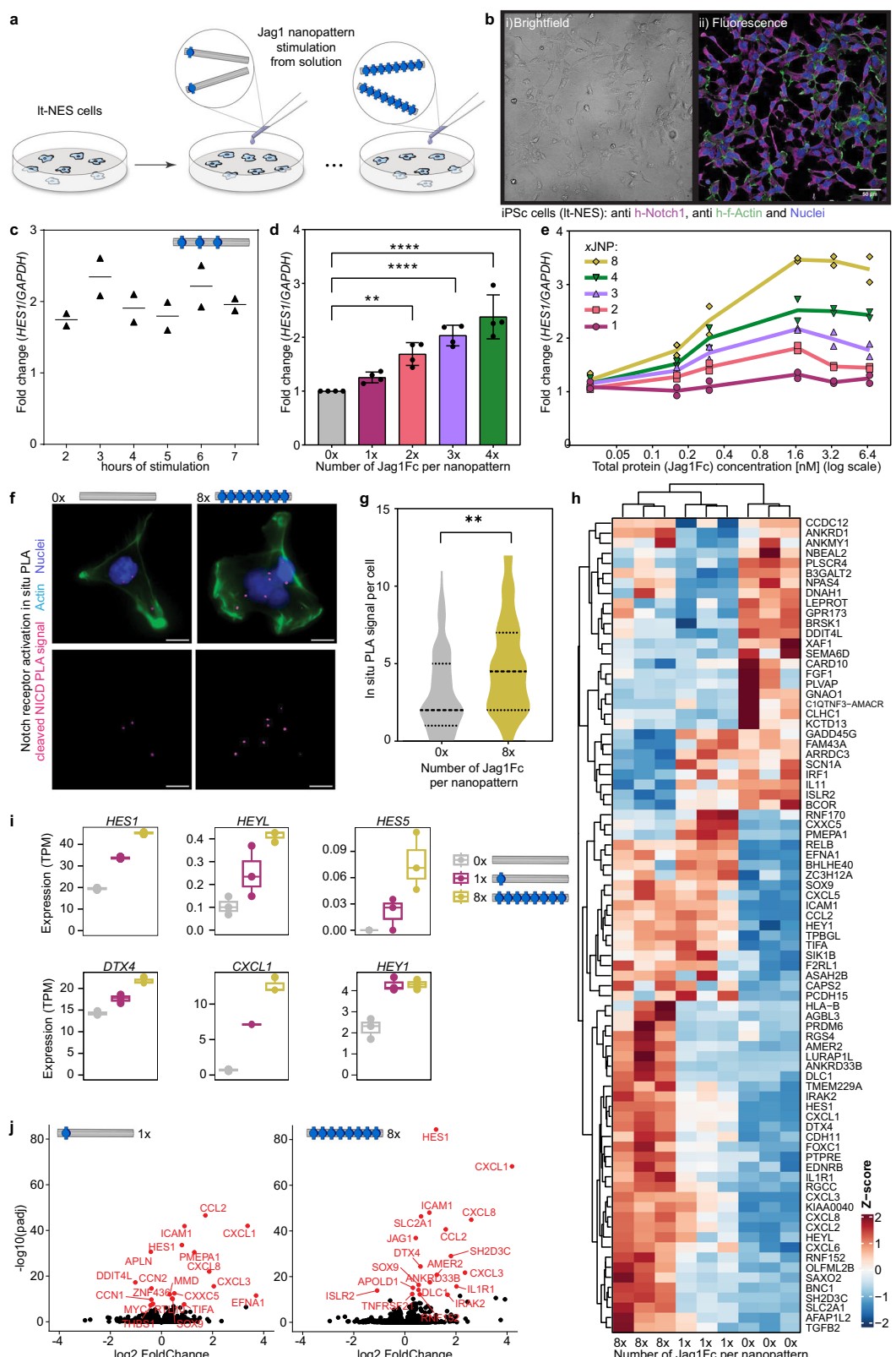

versions of the JNPs in the presence or absence of an inhibitor against Clathrin-mediated endocytosis. We first confirmed that the inhibitor had the desired endocytosis-blocking effect in the lt-NES cells (Methods and Supplementary Fig. 7). When using the inhibitor in stimulation experiments, we observed no change in the multivalency effect (Fig. 4c). Moreover, we did not observe marked differences in the expression of genes known to be involved in endocytic processes

in the RNA-seq experiments (Fig. 4d). It can be interjected that these basal function genes are expected to be reasonably stably expressed. However, nanoparticle uptake has previously been shown to change transcription of endocytosis genes[38], but here we do not observe this. Additionally, endocytosis inhibitors like the one we use here, are not expected to block endocytosis completely (Supplementary Fig. 7). Nonetheless, the fact that we observed that Jag1 multivalency

**Fig. 3 | Activation of the Notch pathway by Jag1Fc-nanopatterns. a** Origami decorated with 1-4 (28 nm separations) or 8 (14 nm separations) Jag1Fc are used to stimulate iPSc-derived neuroepithelial stem-like (lt-NES) cells. **b** Microscopy of lt-NES cells shown in i) brightfield and ii) fluorescence channel immunostained for Notch1 (magenta), F-actin (green), and nucleus (blue). Scale bar, 50 μm. **c** Time course of lt-NES cells stimulated with 3x JNP. Relative expression of *HES1* from qPCR normalized to parallel samples stimulated with empty(0x) structures. Points represent individual data for *n* = 2 biological repeats **d** Effect of different number of Jag1Fcs. Relative expression of *HES1* from qPCR normalized to sample with empty structures for 1-4x JNPs. Bar graphs represent mean expression levels ± SD and black dots indicate individual data points for *n* = 4 biological replicates. Statistical analysis of the data was performed using one-way analysis of variance (ANOVA) followed by Dunnett multiple-comparison test (**P = 0.0026, ****P < 0.0001). **e** Dosage effect. lt-NES cells stimulated with 1-8x JNP at increasing concentrations. Relative expression of *HES1* from qPCR normalized as in (**c**). Points represent individual data for *n* = 2 biological repeats **f** Proximity Ligation Assay (PLA)

performed using antibodies against cleaved NICD on stimulated lt-NES cells. Representative images of cells for each condition: PLA dots (magenta), F-actin (green) and nucleus (blue). Scale bars, 10 μm. **g** Violin plot of the PLA experiment from image analysis of 50 cells for each condition. Statistical analysis of the data was performed using ANOVA followed by Tukey multiple-comparison test (**P = 0.0011). **h** Heat map diagram of mRNA sequencing experiment performed on lt-NES cells after stimulation with JNPs. Three biological repeats for each condition were shown for genes with FDR < 0.05 and absolute log2FC > 0.5. Data is shown automatically clustered using hierarchical complete-linkage clustering of Euclidean distances. **i** Selection of Notch pathway-related genes from RNA seq, transcripts per million (TPM) plotted for each condition. Box plots shown as median, first and third quartiles with whiskers extending up to 1.5 x inter-quartile range (IQR) and individual data points represent data for *n* = 3 biological repeats **j** Volcano plots of genes upregulated by 1x JNP and 8x JNP relative to 0x JNP. Source data are provided as a Source Data file.

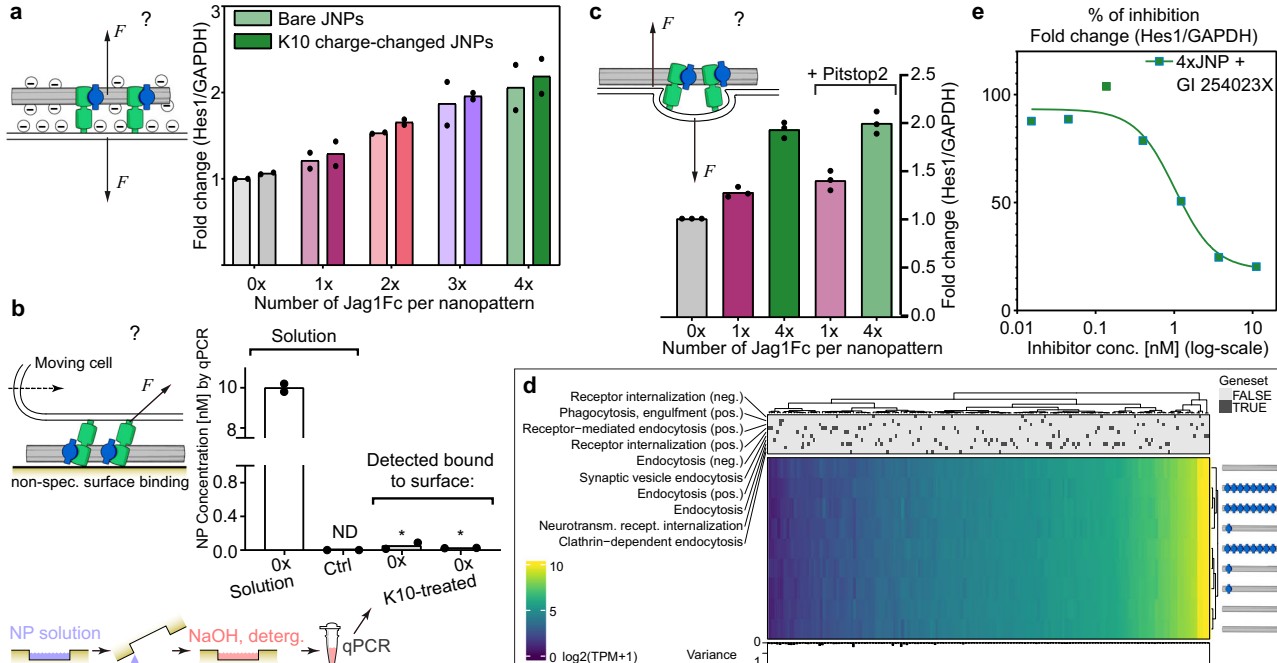

**Fig. 4 | Multivalency drive Notch activation despite potential force sources and ADAM10 is the driving metalloprotease. a** Electrostatic repulsion between negative DNA origami and negative cell membrane could cause pull. qPCR on lt-NES cells stimulated with normal JNPs, or oligolysine (K10) coated JNPs (charge changed towards neutral). Bar graphs show the mean expression levels and dots represent 2 biological repeats. **b** Non-specific surface attachment of JNPs could drive force generation in motile cells. qPCR of part of the DNA origami scaffold (M13mp18 ssDNA) to detect concentrations of structures in solution and, retrieved by harsh washing of cell culture surfaces exposed to JNP solution, any remaining non-specifically bound DNA origami. *Denotes measurements below the linear range of

the assay. Bar graphs show the mean value of 2 technical repeats shown with dots **c** Endocytosis could drive force generation. qPCR of *HES1* with, or without, Pitstop2 (inhibitor for clathrin-mediated endocytosis) on JNP-stimulated lt-NES cells. Bar graphs show the mean value of 3 technical repeats shown with dots. **d** RNA sequencing data of gene ontology genesets that are implied in different levels of endocytosis. Data is shown automatically clustered using hierarchical complete-linkage clustering of Euclidean distances. **e** Increasing concentrations of ADAM inhibitor GI 254023X added on iPS cells and stimulated with 4x JNPs. qPCR of *HES1*. Square symbols represent the mean value of 3 technical repeats. Source data are provided as a source data file.

determined Notch activation levels independently of endocytosis modulation suggests that these responses are not attributed to internalization effects.

Both ADAM10 and ADAM17 have been reported to be able to trigger the release cascade of the NICD but in a context-dependent manner. ADAM 10 appears to be responsible for NOTCH activation induced by ligands while ADAM17 is primarily viewed as responsible for ligand-independent activation[8,39]. Using an inhibitor selective for ADAM10 (GI 254023X, Methods) in our stimulation assay (Fig. 4e) we observe that the IC$_{50}$ concentration for this compound was as low as

1.7 nM (too low for ADAM17 inhibition) showing that ADAM10 is the metalloprotease implied in the activation using JNPs and thus that the activation is ligand dependent[39].

**Chimeric patterns suggest a time-of-binding-dependent effect**
Although the above results suggest that activation of Notch might occur without a pulling force being applied by the ligand, the question remained whether the multivalency effect we observed is due to clustering of the receptor or if the effect is mainly due to avidity and increased time-of-binding ligand-receptor pairs (Fig. 5a). To test

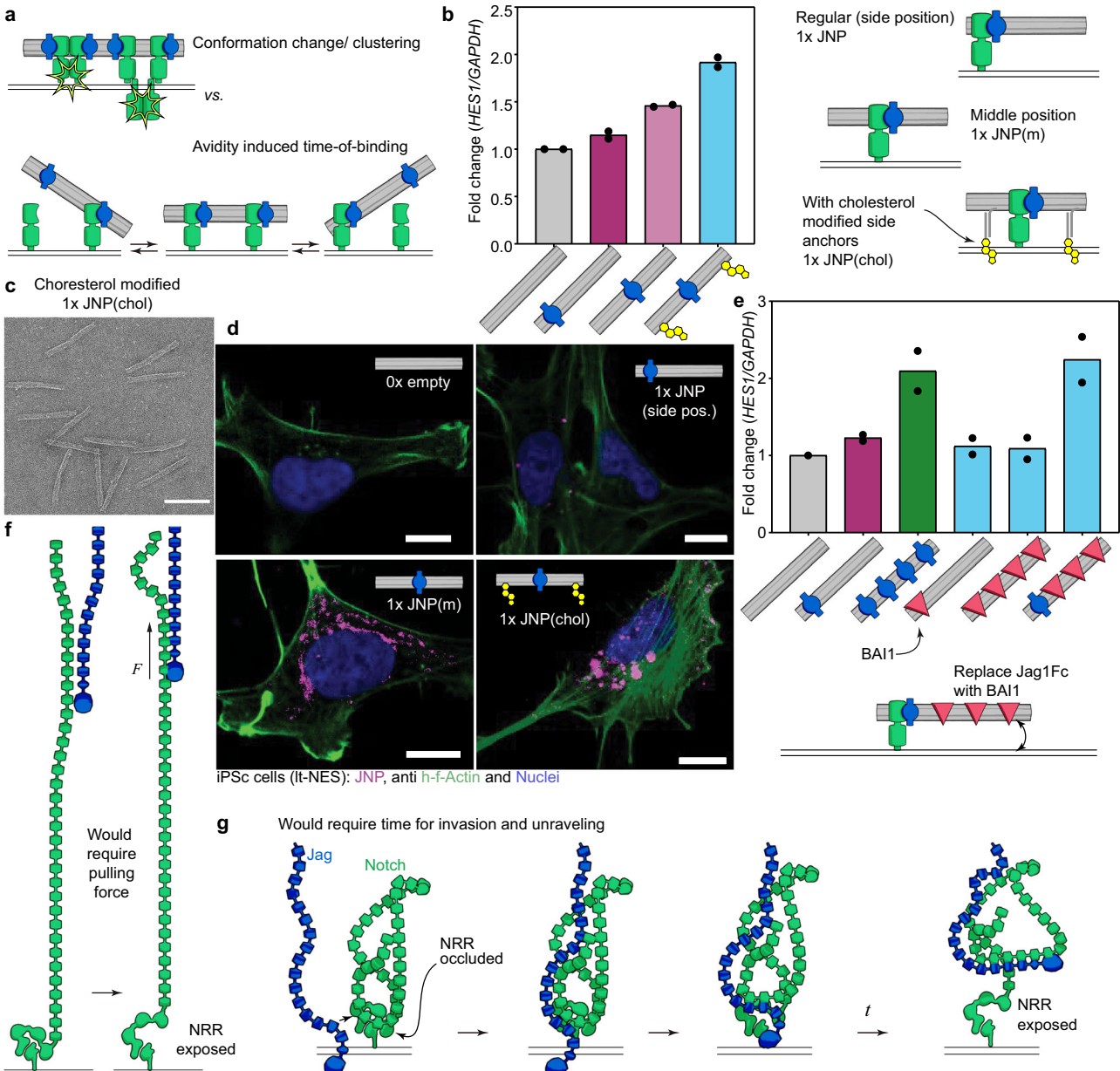

**Fig. 5 | Chimeric structures indicate that multivalency effect is primarily caused by increased binding and suggests hypothesis of tertiary structure occlusion.**
**a** Are the effects observed reliant on receptor clustering and following conformational change (top), or are they due to a reduced off-rate of bound structures leading to more time for activation of single receptors (bottom)? **b** Cholesterol chimeras. Activation data from qPCR of stimulated lt-NES cells (left) and schematic representation (right) of 1x JNP with protein placed at the side as before, in the middle, and in the middle with cholesterol modifications. Bar graphs show the mean value and dots represent *n* = 2 biological repeats. **c** Micrographs of cholesterol-modified JNPs Negative stain TEM. The scale bar is 100 nm. Micrographs repeated independently *n* = 3 **d** Confocal microscopy of stimulated cells

after stimulation with fluorophore labeled 0x JNP, 1x JNP, 1x JNP(m), and 1x JNP(chol). Images repeated independently *n* = 3. Scale bars are 10 µm. **e** BAI1 chimeras – where Jag1Fc is swapped at certain positions with the protein BAI1. Fold change of *HES1* from qPCR in lt-NES cells stimulated with BAI1/Jag1Fc chimeric JNPs. Bar graphs show the mean value and dots represent *n* = 2 biological repeats. **f, g** Different conceptual models of Notch activation. **f** Where EGF repeats of Notch (green) extend out like a rod from the cell membrane (lines at the bottom), a force would be able to transport the signal of binding down to the cell membrane to expose the NRR. **g** Where EGF repeats of Notch are curled up in a tertiary structure, a binding of ligand would be able to induce unraveling of the tertiary structure over time to allow exposure of the NRR. Source data are provided as a Source Data file.

whether simply increasing the avidity of the structures without using more Jag1 ligands, we designed chimeric structures carrying different moieties that should also bind cell surfaces, placed on the same structure as Jag1Fc.

First, we positioned a single Jag1Fc in the middle of the DNA nanostructure and added one cholesterol-modified DNA strand at each side about 70 nm away from the Jag1Fc (Fig. 5b) to act as membrane anchors (Methods). The structures remained monodisperse as could be observed by TEM of the cholesterol-modified structures (Fig. 5c). A

control JNP with the same layout lacking cholesterol was also tested. Incorporation of Cholesterol moieties on the JNP caused higher activation of the Notch pathway, as well as higher apparent binding in microscopy (Fig. 5d), similar to what we previously observed for the 4x JNPs, which suggests that a prolonged residence time is the primary factor helping the complex to accomplish a successful activation.

To further control for the hypothesis that Notch activation could depend on increasing the time of interaction with the ligands, we produced another chimeric DNA nanopattern in which some of the

Jag1 ligands were replaced by BAI1 protein. This protein has known surface targets in our model system (see Methods). IPS cells were stimulated with chimeric Jag1/BAI1 nanopatterns (Fig. 5e and Supplementary Fig. 5b) and we observed that BAI1 protein alone didn't activate the Notch pathway but when it was placed together with 1x Jag1Fc protein on a DNA nanotube the same activation level was reached as with 4x JNP. A further test where the molecular weight is increased without adding molecules that mediate interaction with the cell membrane was performed using the addition of extra DNA helices to the JNPs (Supplementary Fig. 8), in this case, the activation levels did not increase. We note that these experiments further rule out charge-effects and molecular weight-effects being important for the multi-valency effect we observe because the BAI1 proteins used bear similar pKA and molecular weight as the Jag1Fc and the extra DNA helices increases the overall negative charge and introduces extra molecular weight that is almost identical to the molecular weight of 3 added Jag1Fc. Overall, the results of the chimeric structures indicate that the increase in Notch activation with multivalency that we observe is probably due to an increase of binding time at the receptor.

## Discussion

Here we have provided evidence that by using nanoscale control of patterns of Jag1, it appears to be possible to activate Notch from solution even when, according to our controls, pulling forces on the receptor are not expected to be active. That a force-dependent mode of activation could also exist is not contradicted by our data – these modes could be complementary and one could be more dominant. Biophysical and cell-biology studies have provided support for the hypothesis that Notch ligands primarily induce activation by cell-cell contacts and force exertion on the receptor as a consequence[13–16]. Nevertheless, there are also a number of studies suggesting long-range activation (without cell-cell contact) or activation by secreted Notch ligands[18,19]. As well as recent techniques to stimulate Notch, probably by a mechanism similar to what we see here, using ligand coated beads in solution[20] or antibody-clustered Notch ligands[21]. How these observations would be explained using exclusively a pulling-force dependent mode of activation is less clear.

We argue that support for the current, stretched-rod structural model of Notch, which has been used to exclude other mechanisms than force, is incomplete. Without itself being experimentally verified, the stretched-rod model of the Notch ECD has to some extent provided indirect support for the pulling force model: If indeed, the receptor confirmation is that of a stretched rod (Fig. 5f), then it is hard to explain how the crucial region close to the membrane (the Negative Regulatory Region, NRR) would be affected other than via a force signal traveling down the length of the receptor. Because the ligand-binding domain is far from the membrane along the primary sequence (Fig. 5f), a pull on the extremity would be required to induce a conformation change on the distant NRR. While the structure is well characterized in terms of the ligand binding domain[11] and the NRR[7], a complete picture of the extracellular domain of Notch receptors remains unsolved, and data from pairs of EGFs supports the existence of mostly flexible joints between them[40]. If the EGF repeats act more as beads on a string which appears to be the case, the biophysical concept of an entropic spring would actually favor a condensed-, as opposed to a stretched-, structure, even without intramolecular interactions. One of the few attempts at looking at the entire ECD of Notch with electron microscopy shows bundled-up structures whose size would be incompatible with a stretched rod model[41].

If the Notch ECD is curled up in a tertiary structure like this suggests (Fig. 5g), other types of feedback from the binding domain to the NRR could be possible. Our data on activation induced by Jag1 nanopatterns and chimeric patterns favors an explanation where more activation is induced by reducing the off-rate of the structures by increased binding to the membrane via either more Jag1, or other

molecules (cholesterol, Bai1). This suggests a slower mode of activation that could be another way of exposing the NRR region for ADAM10 cleavage. Similar to DNA strand invasion[42] (a slow reaction where a random walk of bound bases eventually leads to the release of a bound DNA region) the Jag1 could by its binding, invade a ravel of Notch EGFs that eventually leads to the release of material from the NRR and its subsequent exposure (Fig. 5g). Note that in this model, a pulling force exerted on the ligand, would probably act as a catalyst for the reaction as the pulling away of bound EGF regions would speed up the exposure of the NRR. In that sense, this hypothesis would stand as a complement to the force model, and not necessarily as a contradiction. This view of a more complex interaction between the Notch ECD and its ligands has been suggested before, after studying deletion mapping and binding[43], antibody binding interference[44] and, more recently, cross-linking mass spectrometry of Notch[45]. But that this type of binding could lead to a slower, force independent mode of activation has not previously been suggested.

Another alternative activation mechanism that is also compatible with our data would be something akin to a kinetic segregation model[46]. In this model for T cell activation a balance between kinase and phosphatase is disturbed when the bulky CD45 phosphatase is assumed to be excluded from the close synapse forming upon T cell receptor binding, thus toppling the balance in favor of the CD79 kinase which then initiates the signaling. I.e. opposite to clustering, a segregation of molecules would lead to activation. It is possible that the presence of the bulk of a 5 MDa DNA origami nanostructure, could be enough to disturb the local balance between ADAM10 and some regulatory counterpart that gets sufficiently excluded under the origami to facilitate the activation. Our results where placing a single Jag1Fc in the middle of a nanostructure, as opposed to one of the ends, lead to slightly higher activation, fit less well with the time-of-binding hypothesis (Fig. 5b), and might be related to this model. That ADAM10 could rely on other regulatory factors has been suggested before[9] we suggest that these might be inhibitory, and their exclusion required for proper cleavage.

Due to the fact that we are using Jag1Fc and not monomeric Jag1 binding domains, we cannot exclude that dimerization of Notch receptors are important for activation. However, the results concerning multiple Jag1Fc patterns, combined with the results from chimeric structures, point to an effect of multivalency, leading to an increased time of binding, as opposed to an effect of receptor clustering induced by the JNPs (because if the latter is the requirement, why do the chimeric structures increase activity?). Future experiments similar to what we present here where a suitable production of monomeric Jag1-DNA conjugates is developed, could provide an answer to whether dimerization is in fact needed as a minimum for activation.

It is interesting to compare our results with studies on synthetic Notch receptors. In SynNotch receptors[22], only the transmembrane- and regulatory- (NRR) regions are kept intact, whereas the rest of the Notch receptor, both the ECD and ICD are replaced by artificial binder regions and transcription regulators, respectively. These artificial Notch-like constructs are assumed to require cell-cell contact for activation, which is explained with a force model. Interestingly, a recent study investigating differences between SynNotch and WT Notch found that SynNotch, as opposed to WT Notch, does not require an intracellular domain on the ligand cell side to initiate activation[23]. Another type of synthetic Notch, called SNIPR, has recently been introduced where the NRR is replaced altogether[24]. When comparing the activation of these systems with our data on endogenous Notch, a complex picture arises. In SynNotch, the EGF domains are replaced, but the activation is still dependent on ADAM10 and γ-Secretase. Although we did not test SynNotch with our nanopatterns, one could argue that these earlier results, taken together with our data, appear to favor some type of kinetic segregation model as an explanation, as the EGF-unraveling hypothesis would not apply to these artificial Notch

mimics. In particular the SNIPR-constructs, which appear to be both dependent on ADAM10 and γ-Secretase for activation despite sometimes lacking the NRR altogether, would be difficult to explain with a force model (as well as with the EGF-unraveling model) because in this theory the NRR region acts as a switch that only reveals the S2 cleavage site if pulled upon.

In addition to providing insight to the basic mechanisms of Notch activation, the results we present here lay the foundation for an alternative development strategy for new soluble Notch agonists. These are currently an elusive and highly sought-after class of drug. Attenuation of Notch signaling due to tumor growth has been shown to cause immunosuppression that could be overcome by enhancing Notch activation in the hematopoietic microenvironment[21]. Targeted Notch activation can also be beneficial in several other diseases: it inhibits acute myeloblastic leukemia growth and increases survival[47], it is suggested as a treatment for Notch ligand loss of function diseases like Alagille Syndrome and for regenerative medicine[48,49].

In conclusion, using well-defined molecularly precise patterns of Jag1 ligands we provide evidence that the Notch pathway can be activated from a solution phase in a manner that is not dependent on a force activation mechanism. Instead, we show that by increasing the number of Jag1 ligands, we increase the activation efficiency. The fact that this effect remains even when one Jag1Fc is combined with other binders (cholesterol, Bai1) makes it difficult to explain this effect through a clustering model. This leads to two possible conclusions: (i) either increased avidity and the time of binding at the receptor is enough to initiate the pathway, potentially via an unraveling of the EGF domains from the NRR, or (ii) the bulk of the nanostructure together with long enough binding time, mimics a neighboring cell in a way sufficient to modulate a kinetic segregation model of unknown players, most likely related to ADAM10.

## Methods

### Design of DNA origami nanostructures
The design of the 18 helix-bundle nanorod was done using caDNAno [https://cadnano.org] using hexagonal lattice design parameters for 3D DNA origami described in ref. 50. The origami design follows the one used in ref. 28 and has been uploaded to a repository [https://nanobase.org] where the design files and structure files can be downloaded. By folding the structure either with 5' protruding staples, or empty site staples, respectively, placement of DNA-protein conjugates can be done at will along the top edge of the structure as described in ref. 28.

### Folding DNA nanostructures
The standard folding conditions used in this study were as follows: 20 nM ssDNA scaffold, 100 nM per staple, 13 mM MgCl2, 5 mM Tris, pH 8.5. Single-stranded DNA that serves as scaffold for DNA nanostructures folding was produced, extracted and purified from M13 phage variant p7560 cultured by inoculating E. coli JM109. The approximately 200 single stranded DNA oligonucleotides helper strands (staples), were purchased from Integrated DNA Technologies. Folding was carried out by annealing at 65 °C for 4 min, then 65 °C to 50 °C for 1 min/0.7 °C, 50 °C to 35 °C for 1 h/1 °C and 20 °C forever until retrieved. Removal of excess staples was done by washing (repetitive concentration and dilution for seven times) with PBS, pH 7.4, 10 mM MgCl2 in 100-kDa MWCO 0.5-ml Amicon centrifugal filters (Millipore). Samples were diluted to 450 μl and transferred to a prewetted centrifugal filter and centrifuged at 10,000 g, for 1 min, and then diluted again to 450 μl, mixed well and centrifuged again under the same conditions. Sample was collected via inverting the filter on an Eppendorf tube and centrifugation at 1000 g for 2 min.

### Jag1 protein production
Jag1 plasmid was a gift by Susan Lea[10]. Jag1Fc protein was produced in human embryonic kidney 293 T (HEK293T) cells by using transient transfection. Plasmid and transfection reagent (lipofectamine 2000) were mixed at a ratio 1:3 in optimem media and added into cells. After 1 day media collected and replaced with 10% FBS DMEM and cells were let to produce proteins for 3 more days. Proteins containing His tag at the C terminus were purified with affinity purification column His trap FF.

### Protein conjugation to DNA
Largely following the method for site-specific labeling at His-tags outlined in[51]. Proteins containing 6x Histidine at the C terminus reacted with the chemical Bis-sulfone-DBCO for 4 h at room temperature. The bis-sulfone group reacts with the histidines at the C terminus of the protein, and then an azide modified oligonucleotide was conjugated to the protein by click chemistry between the azide-and the DBCO-group. In the case of Jag1 protein, we used the sequence: CTCTCCTTCTTCCCTTTCTTT while in the case of BAI1 protein we used the sequence: TTCGACAGCATGAACATCAGC.

### Jag1Fc nanopatterns
The ligand Jag1 conjugates were added with a twenty times excess to each protruding site on the DNA nanostructure and incubated in a PCR machine with a temperature ramp starting from 1 h at 37 °C followed by 14 h at 22 °C, and immediately after incubation the nanopatterns were stored at 4 °C. Removal of Jag1 conjugates in excess was performed in an FPLC system with a size exclusion purification column (Superose 6 Increase 10/300GL, Cytiva). After purification, fractions of the peak corresponding to Jag1 nanopatterns were collected and concentrated in 30-kDa MWCO 2-ml Amicon centrifugal filters (Millipore). Sample concentration was estimated with agarose gel electrophoresis by loading samples of known concentration before purification and samples after purification of unknown concentration. By comparing the intensity of the bands we calculate the final concentration of each sample.

### Agarose gels for characterization of the Jag1 nanopatterns
We prepared 2% agarose gels with 0.5× TBE supplemented with 11 mM MgCl2 (Sigma-Aldrich) and 0.5 mg/ml ethidium bromide (Sigma-Aldrich). We typically loaded 4 μl of 20 nM DNA nanostructures in each lane and ran the gels in 0.5× TBE with 10 mM MgCl2 at 90 V for 3 h, cooled in an ice-water bath. The gels were imaged on a GE Image Quant LAS 4000 system.

### Transmission electron microscopy (TEM)
We applied 3 μl of the DNA nanostructures on glow-discharged, carbon-coated Formvar grids (Electron Microscopy Sciences), incubated for 20 s, blotted off with filter paper, and then stained with 2% (w/v) aqueous solution of uranyl formate supplemented with 20 mM NaOH followed by a final blot with filter paper. The negatively stained samples were imaged by Talos 120 V microscope at x92k magnification. To visualize the internal helices of the 18-helix bundle (18HB) with additional DNA sequences (0x JNP + DNA) and compare them with the original 18HB (0x JNP), we employed negative stain electron tomography. Grid preparation followed the previously described protocol (5 nM concentration, 1.5 min incubation, and 40 s staining). Tilt series were captured from −40° to 40° at 1° increments using a magnification of x57k. These series were then processed using the Etomo program from the IMOD package[52], resulting in the generation of tomograms.

### Surface plasmon resonance (SPR)
A Biacore T200 (GE Healthcare) was used to measure the binding kinetics of Jag1Fc nanopatterns to Notch receptor. Streptavidin (Sigma-Aldrich) was dissolved in 100 mM sodium acetate buffer, pH 4.5, and immobilized on a CM3 chip (GE Healthcare) according to the manufacturer's instructions. The biotinylated extracellular domain of the human Notch1 receptor (EGF 8–12)[11] was immobilized at 200 RU.

Jag1Fc DNA nanopattern samples were diluted to concentrations ranging from 2,5 nM to 10 nM in PBS, pH 7.4, supplemented with 10 mM MgCl2. The flow rate of the samples was adjusted to 5 µl/min, and a total amount of 35 µl was injected. Sensorgram data were processed with BIAevaluation 3.2 software (GE Healthcare).

### Sample preparation for DNA PAINT imaging experiments

Versions of Jag1 nanopatterns (JNPs) with one, two, three, and four Jag functionalization sites used in other experiments were produced carrying biotinylated anchoring-sites on the opposing side to the Jag sites, along with six internal Cy5 modified staple-oligonucleotides for DNA PAINT-independent detection of the nanopatterns. Nanopatterns were produced and purified as described earlier using Jag1 proteins conjugated to oligos containing the anchoring sequence and two DNA PAINT docking sequences. Microscope slides (VWR) and coverslips (1.5H, VWR) were cleaned with acetone and isopropanol and flow-chambers were produced by placing two strips of double-sided scotch tape approximately 0.8 cm away from each other on the slides and placing the cleaned coverslips on top of the strips. The channel was first incubated with biotinylated-BSA (Sigma Aldrich)) solution (1 mg/mL biotinylated-BSA in Buffer A (10 mM Tris-HCl, 100 mM NaCl, 0.05% Tween-20, pH 7.5)) for 2 min and then washed with Buffer A. The channel was then incubated with streptavidin (Thermo Scientific) solution (0.5 mg/mL streptavidin in Buffer A) for 2 min. Following a washing step with Buffer A the channel was washed with Buffer B (5 mM Tris-HCl, 10 mM MgCl2, 1 mM EDTA, 0.05% Tween-20, pH 8). The channel was then incubated with Jag- nanopatterns solution (50 pM Jag-NC in Buffer B) for 5 min followed by washing the channel with Buffer B. Finally, the imager-solution (10 nM Atto550-labeled imager strand in Buffer B+ (Buffer B, oxygen scavenger system (2.4 mM PCA (Sigma Aldrich) and 10 nM PCD (Sigma Aldrich)) and 1 mM Trolox (Sigma Aldrich)) was introduced into the channel and the channel was then sealed with epoxy glue.

### DNA PAINT imaging of Jag-functionalized nanopatterns

The DNA PAINT imaging experiments were conducted with a Nikon Eclipse Ti-E inverted microscope (Nikon Instruments) using a 1.49 NA CFI Plan Apo TIRF 100× Oil immersion objective (Nikon Instruments) and a 1.5x auxiliary Optovar magnification. The TIRF illumination was produced using an iLAS2 system (Gataca systems) with an OBIS 561 nm LS 150 mW laser (Coherent) and an Omicron LuxX+ 642 nm 140 mW laser (Coherent) and a custom input beam expansion lens (Cairn). The excitation light was filtered with a filter cube (89901, Chroma Technology), an excitation quadband filter (ZET405/488/561/640x, Chroma Technology) and a quadband dicroic (ZET405/488/561/640bs, Chroma Technology). The emitted light was first filtered with a quadband emission filter (ZET405/488/561/640 m, Chroma Technology) and an additional emission filter (ET595/50 m, Chroma Technology; ET655lp, Chroma Technology) and the signal was recorded with an iXon Ultra 888 EMCCD camera (Andor) using the Micromanager software. For the recording of the Cy5 signal snapshots were taken with 1 s exposure time, 10 MHz readout rate and no EM. For the recording of the DNA PAINT data 12000 frames were collected in frame-transfer mode with 300msec exposure time, 10 MHz readout rate and no EM gain.

### Quantification of Jag proteins on nanopatterns using DNA PAINT

**Fitting of localizations.** The Picasso software package[53] was used for preprocessing the raw DNA PAINT data. The Picasso Localize software was used to detect and fit localizations in the raw DNA PAINT movies using the MLE algorithm (Box sixe:7, Min. Net Gradient: 2000, EM Gain: 2, Baseline: 43.2, Sensitivity: 4.1, Quantum efficiency: 0.98, Pixel size: 87 nm).

**Drift correction and filtering of DNA PAINT data.** Following fitting the Picasso Render software was used to drift correct the localizations

using the redundant cross-correlation (RCC) algorithm (segment size: 200 frames). The data was subsequently filtered using a custom Python script through the removal of low precision localizations and multi-event localizations (Supplementary Table 1). The filtered localizations were drift corrected again using the Picasso Render software: individual origami structures were picked using the pick similar tool (pick area diameter:1.5 camera pixel, pick similar std: 1.6), initiated with 20 manually picked origami structures, and the data was undrifted with the undrift from picked feature of the software.

**Detection of DNA origami.** Detection of DNA origamis were performed with a custom Python script. The Cy5 image together with the DNA-PAINT data was used to determine the position of individual DNA origami probes. First the Cy5 image was intensity normalized and a binary image was produced with adaptive thresholding. After noise removal the image was segmented, and the contours were detected for the individual segmented objects. The DNA PAINT data was used in parallel to generate a low-resolution image from where the positions of DNA origami probes were determined by generating a pixel-inflated binary image and detecting the contours of objects in a selected size range. The final positions of origami probes were generated from the two set of contour coordinates (Cy5 and PAINT) by combining them: in the case of Cy5 contours with overlapping DNA PAINT contours (distance between contour centers smaller than 90% of sum of contour radiuses) the contour coordinates generated from the PAINT data were used, in the case of Cy5 contours with no overlap (structures without detected Jag1 proteins) the Cy5 contour coordinates were used. Using the center coordinates of these contours and the average size of DNA PAINT contours coordinates for the origami region of interests (ROIs) were determined and DNA PAINT localizations were grouped into these. (Supplementary Fig. 1b).

**Quantification of Jag proteins within DNA origami ROIs.** Reference values used in the later processing steps for neighboring position-to-position distances and linearity scores were calculated using the custom Python script. For the calculation of the position-to-position distances the localizations grouped into individual origami ROIs in the 2xJNP dataset were clustered using the DBSCAN algorithm and the distance between the mean position of the two clusters with the highest number of points in them was extracted. The reference value for the position-to-position distance was then calculated as the mean of the gaussian fit of the resulting distance distribution. (Supplementary Fig. 1c). For calculating the linearity score cut-off value, the localizations in individual ROIs in the 4xJNP dataset were rendered into a high-resolution image with intensity normalization to the highest pixel value. Localization density maxima were detected as protein positions and after removing outlier points the protein positions were annotated by using their distance matrix. Position to neighboring position (PTNP) vectors were calculated between the adjacent protein positions and the mean standard deviation of the normal PTNP vectors' x, y coordinates was calculated as the linearity score for each probe. The cut-off value was then determined as the inflection point of the cumulative distribution of this linearity score for origami ROIs with four detected points (Supplementary Fig. 1d).

Quantification of Jag proteins on individual origami probes residing in the origami ROIs was performed using a custom Python script. Localizations in individual origami ROIs were rendered into a high-resolution image with intensity normalization to the highest pixel value. Initial guesses for the protein positions were determined by detecting localization density maxima in the image and annotating them using their distance matrix after the removal of outliers. ROIs with higher linearity score as the cut-off value and/or with more detected positions than the designed were discarded. In ROIs passing this filtering the initial guess for Jag1 position one along with the position-to-position reference distance was used to calculate the

putative regions for each position and these regions were then scanned for local localization maxima to detect density maxima with lower number of localizations. For positions with detected maxima the coordinates of these were then used as the final positions of Jag1 proteins. (Supplementary Fig. 1e).

## Culture of Neuroepithelial Stem (NES) cells AF22

lt-NES samples were obtained from the iPS Core facility at Karolinska Institutet. NES cells[30,54] were cultured as adherent cells in cell culture flasks previously coated with 20 µg/ml polyornithine (Sigma) for 1 h and 1 µg/ml Laminin2020 for 4 h (Sigma). Cells were cultured on NES culture medium contained DMEM/F12+GlutaMax (Gibco), supplemented with 10 µl/ml N-2-supplement (100×, Thermo Fisher Scientific), 10 µl/ml Penicillin-Streptomycin (10,000 U/ml, Thermo Fisher Scientific), 1 µl/ml B27-supplement (50×, Thermo Fisher Scientific), 10 ng/ml of bFGF (Life Technologies) and 10 ng/ ml of FGF (Pepro-Tech). The culture medium was replaced every second day. The NES cells were passaged enzymatically when reaching 100% confluency using Trypsin-EDTA (0.025%, Thermo Fisher Scientific). NES cells were seeded at a density of 40,000 cells/cm$^2$.

## Stimulation of Nes cells with Jag1 nanopatterns

Nes cells were seeded at the density of 18750 cell/cm$^2$ and let inside the cell culture incubator to attach for 6 h. Cells were stimulated with different Jag1Fc nanopatterns for 3 h when we performed either RNA extraction for real time q-PCR and mRNA sequencing experiment or fixed for proximity ligation and other microscopy experiments.

## Inhibitor experiments

In the study we used known inhibitors of the Notch signaling pathway such as: γ-secretase inhibitor DAPT (Sigma-Aldrich, D5942), used at 10 µM; a general matrix metalloproteinase (MMPs) inhibitor, Batimastat (Sigma-Aldrich, SML0041), used at 10 µM. We also used a clathrin mediated endocytosis inhibitor, Pitstop2 (Sigma-Aldrich, SML1169), used at 25 µM. In each case, these inhibitors were added to the cells 2hrs prior to stimulation. To further validate the successful inhibition of clathrin mediated endocytosis by Pitstop2, we added 40 µg/ml Transferrin from human serum Alexa Fluor 594 (Thermo Fisher Scientific, T13343) to the cells at 37 °C for 20 min. Then cells were fixed with 4% formaldehyde at 37 °C for 15 min, washed with PBS and stained with NucBlue Fixed Cell ReadyProbes Reagent (Thermo Fisher Scientific, R37606). To assess whether Jag1Fc nanopatterns do induce ligand dependent activation, we used a selective inhibitor for ADAM10 (GI 254023X (Sigma-Aldrich, SML0789)) in our cell culture experiments. The GI 254023X inhibitor has a 100-fold selectivity for ADAM10 over ADAM17. In our experiment, we added increasing concentrations of GI 254023X inhibitor, 2 h prior to cell stimulation with 4x JNPs. The relative inhibition, for samples with increasing concentration of inhibitor, was calculated relative to a sample without added inhibitor (Fig. 4e) and the IC50 (half maximal inhibitory concentration) of the compound was calculated to be 1.7 nM.

## RNA extraction

RNA extraction for samples used in RT-qPCR experiments was performed using the Cells-to-CT kit (A25599, Thermo Fisher) according to manufacturer instructions. In most of the experiments, 6000 cells were seeded in a 96-well plate. RNA extraction for samples used in RNA-sequencing experiments was performed by using RNeasy Micro Kit (Qiagen).

## Real time-qPCR

cDNA and RT-qPCR experiments were performed according to the instructions included on the Cells-to-CT kit. The pcr reaction mixture includes 10 µl of SYBR green, 1 µl of primers (250 nM final concentration), 5 µl of water, and 4 µl of cDNA. Primers used in this study are

*HES1* Fw: AGG CGG ACA TTC TGG AAA TG *HES1* Rev: TCG TTC ATG CAC TCG CTG A *GAPDH* Fw: ACT TCA ACA GCG ACA CCC ACT *GAPDH* Rev: CAC CCT GTT GCT GTA GCC AAA.

## Proximity Ligation Assay

We applied a proximity ligation assay[55] with two secondary antibodies targeting a primary antibody directed to the epitope of Notch that is revealed on the released NICS only after γ-secretase cleavage, similar to[56]. After stimulation, cells were fixed with 4% final concentration of methanol-free formaldehyde (Thermo Fisher Scientific, cat. no. 28908) for 15 min at 37 °C. The cells were washed at room temperature three times for 5 min each with 1x PBS (Sigma Aldrich, cat. no. 806552). Cells were then permeabilized with 1x PBS 0.1% Triton X-100 (Sigma Aldrich, cat. no. 93443) for 15 min at room temperature and washed at room temperature three times for 5 min each with 1x PBS.

**PLA protocol.** The samples were blocked with Duolink Blocking Solution (Sigma Aldrich, cat. no. DUO92002) for 1 h in a pre-heated humidity chamber at 37 °C. The rabbit IgG monoclonal antibody for the detection of cleaved Notch1 (Val1744) (D3B8) (Cell Signaling Technology, cat. no. 4147) was diluted 1:200 in 1x Duolink Antibody Diluent (Sigma Aldrich, cat. no. DUO92002). The cells were incubated with the primary antibody overnight at 4 °C and washed at room temperature three times for 5 min each with 1x Duolink In situ Wash Buffer A (Sigma Aldrich, cat. no. DUO82049). The Duolink In Situ PLA Probes anti-rabbit PLUS (Sigma Aldrich, cat. no. DUO92002) and MINUS (Sigma Aldrich, cat. no. DUO92005) were then diluted 1:5 in Duolink Antibody Diluent and incubated with the sample in a humidity chamber for 1 h at 37 °C. The probes were washed with 1x Duolink In Situ Wash Buffer A three times for 5 min each at room temperature.

Next, the Ligase was diluted 1:40 in 1x Ligation Buffer (Sigma Aldrich, cat. no. DUO92008) and incubated in a humidity chamber for 30 min at 37 °C. The ligation solution was washed from the cells three times with Wash Buffer A for 5 min each at room temperature. The Polymerase was diluted 1:80 in 1X Amplification Buffer (Sigma Aldrich, cat. no. DUO92008) and amplification of the rolling circle amplification product was carried out in a humidity chamber for 100 min at 37 °C. The cells were washed three times for 10 min each at room temperature with 1x Duolink In situ Wash Buffer B (Sigma Aldrich, cat. no. DUO82049) followed by two washes of 2 min with 0.01x Duolink In Situ Wash Buffer B.

The F-actin of the cells was fluorescently labeled with Alexa Fluor™ 488 Phalloidin (Thermo Fisher Scientific, cat. no. A12379) according to the manufacturer's instructions. Nuclei were stained with DAPI solution (Abcam, cat. no. ab228549) diluted to a final concentration of 2 µM in 1x PBS and incubated for 40 min at room temperature. Finally, the cells were washed three times for 5 min each at room temperature with 1x PBS.

**Imaging and image analysis.** Cells were imaged in 1x PBS with a Nikon Eclipse Ti-E inverted microscope (Nikon Instruments) using a 1.49 NA CFI Plan Apo TIRF 100× Oil immersion objective (Nikon Instruments). The sample was illuminated at a low angle of 15° using the iLAS2 system (Gataca systems) with lasers listed later (Supplementary Table 2) using a custom input beam expansion lens (Cairn). The excitation light was filtered with a filter cube (89901, Chroma Technology), an excitation quadband filter (ZET405/488/561/640x, Chroma Technology) and a quadband dichroic (ZET405/488/561/640bs, Chroma Technology). The emitted light was first filtered with a quadband emission filter (ZET405/488/561/640 m, Chroma Technology) and additional respective emission filter described later (Supplementary Table 2) (Chroma Technology) and the signal was recorded with an iXon Ultra 888 EMCCD camera (Andor) using the Micromanager software with camera parameters describe later (Supplementary Table 2). Each condition was tested with two biological replicates consisting of 50 different cells

imaged with z-stacks containing 10 planes with a step size of 4 µm. The z-stacks of the in situ PLA signal were converted into Maximal Intensity Projections (MIP) using Fiji. From the z-stacks of the F-actin and the nuclei the focused slice was automatically found using Fiji, the images were then pre-processed by adjusting the contrast and brightness and applying a Gaussian blur to improve object detection by thresholding. The nuclei and boundaries of the cells and PLA signals were identified using batch processing with CellProfiler (www.cellprofiler.org).

### Immunostaining of Notch1 receptor

Cells were seeded for 6 h at a density of 75000 cells/cm$^2$ and fixed with 4% methanol-free formaldehyde for 15 min. Then cells were permeabilized with 0.1% Triton™ X-100 in PBS for 10 min, washed twice in PBS, and blocked for unspecific binding of antibodies with 3% BSA for 30 min. Cells were probed with NOTCH1 Monoclonal Antibody (Thermo Fisher SCIENTIFIC, # MA5-11961) in 3% BSA at a dilution of 1:100 and incubated overnight at 4 °C in a humidified chamber. The next day, cells were washed with PBS and incubated with 1 µg/mL of Goat anti-Mouse IgG (H + L) Cross-Adsorbed Secondary Antibody, Alexa Fluor™ 488 (Thermo Fisher SCIENTIFIC, #A-11001) at room temperature in the dark for 1 h. Lastly, we washed the cells with PBS and stained F-actin and nucleus with Texas Red™-X Phalloidin (Thermo Fisher SCIENTIFIC, #T7471) and NucBlue™ Fixed Cell ReadyProbes™ Reagent (Thermo Fisher SCIENTIFIC, #R37606), respectively, via following the product protocols. Cells were imaged on Zeiss LSM980-Airy2. Post-processing of images was done with Fiji.

### Confocal imaging of Cy5-labeled DNA nanostructures

Cells seeded and stimulated as described above. Cy5- labeled Jag1 nanopatterns used at 1.66 nM final concentration. Cells were fixed with 4% methanol-free formaldehyde for 15 min and permeabilized with 0.1% Triton™ X-100 in PBS for 15 min. Then cells were washed with PBS and stained for F-actin and nucleus with Alexa Fluor™ 488 Phalloidin (Thermo Fisher SCIENTIFIC, #A12379) and NucBlue™ Fixed Cell ReadyProbes™ Reagent (Thermo Fisher SCIENTIFIC, #R37606), respectively, via following the product protocols. Cells were imaged on Zeiss LSM980-Airy2. Post-processing of images was done with Fiji.

**Libraries for sequencing experiment.** Libraries for RNA sequencing were prepared with TruSeq Stranded mRNA (Illumina 20020595) according to manufacturer instructions. Libraries were sequenced on an in-house NextSeq550 using high-output v2 kits.

**RNA-seq analysis.** Raw data was adapter trimmed and human transcriptome (GRCh38.p13 Gencode v38 protein-coding transcripts; gencode.v38.pc_transcripts) was quantified using Salmon[57] (v1.3.0, options: -l ISR --validateMappings). Quantifications were summarized to gene-level using tximport[58] (v1.12.3) and differential expression was calculated as Wald tests using DESeq2[59] (v1.24.0). Gene ontology enrichment was performed as Fisher overrepresentation tests using PANTHER[60] (release 20210224, GO database 10.5281/zenodo.5080993 and release 20221103, GO:0030100 Regulation of Endocytosis).

### Statistical analysis

For multiple comparison analysis in Fig. 3d, we performed one-way ANOVA followed by Dunnett multiple comparison. Using Prism software (GraphPad), we performed the Shapiro-Wilk normality test, which showed that each group of data followed a normal distribution. For the in situ PLA statistical analysis of the processed images (Fig. 3f and Supplementary Fig. 4c), one-way ANOVA followed by Tukey post hoc test was performed for multiple-comparison analysis of the four populations comprising the two biological repeats for each condition (Supplementary Table 3). Analysis was conducted in a blinded format whereby knowledge of which conditions applied to which groups of cells held by one author was withheld from the author responsible for

image acquisition and analysis to avoid bias. After visual inspection of the batch processing, cells that were partially out of the field of view (i.e. cut off) were not included in the analysis.

### Oligolysine coating of DNA origami nanostructures and JNPs

Oligolysine (K10, Alamanda Polymers) coating has previously been used to neutralize DNA origamis and protect them from low salt denaturation and nuclease mediated degradation, we followed the protocol in the original publication[61]. Briefly, screening different K10 concentrations relative to origami concentration we found that at a ratio of 0.5:1 nitrogen to phosphorus groups (N:P) in Lysine:DNA, the nanostructures were coated with K10 without forming aggregates which also led to a significantly changed charge towards positive as could be seen from gel electrophoresis and Zeta potential measurements, which were performed on a dynamic light scattering analyzer (Malvern Nano ZS, Malvern) (Supplementary Figs. 5a and 6). The appropriate amounts of DNA origami and K10 solution were mixed and incubated at room temperature for 30 min prior to using them in the downstream experiments.

### qPCR to detect trace concentrations of JNPs on cell culture surfaces

The method to measure DNA origami via qPCR is described in[62], we followed the original publication in terms of protocol and primer design. In our assay we first exposed cell culture surfaces to a solution of typical concentration of JNPs, followed by incubation and then removal of the JNP solution. These cell culture wells were then washed using harsh conditions (50 µl of 8 mM NaOH per well) to extract remnant, potentially non-specifically bound structures, and the recovered washing solution was subject to qPCR directed to the M13mp18-derived ssDNA scaffold of the DNA origamis as described in[62].

### Fabrication of BAI1- and Cholesterol-chimeric structures

To form the cholesterol chimeric structures we hybridized Jag1 conjugates to the DNA origami, the excess of Jag1 conjugates was removed and in a second reaction we added cholesterol-modified oligonucleotides (Chol- TTCGACAGCATGAACATCAGC) in a 1:1 ratio to each protruding site of the DNA origami. Cholesterol contains a hydrophilic hydroxyl group while the tail of the molecule is hydrophobic. Attaching DNA-cholesterol conjugates on DNA origami nanostructures is an established way to force them to attach to lipid membranes[63].

To image cholesterol-chimeric JNPs with the lt-NES cells we introduced 15 Cy5 fluorophores (magenta in Fig. 5d) on the inside of the DNA nanotube structures, using a Cy5 oligo complementary to extended staples that were designed to protrude to the inside cavity of the 18HB origami. After 3 h of stimulation, we fixed the cells, stained for actin (green in Fig. 5d) and the nucleus (blue in Fig. 5d), and imaged the sample with the confocal microscope. The F-actin of the cells was fluorescently labeled with Alexa Fluor™ 488 Phalloidin (Thermo Fisher Scientific, cat. no. A12379) according to the manufacturer's instructions. Nuclei were stained with DAPI solution (Abcam, cat. no. ab228549) diluted to a final concentration of 2 µM in 1x PBS and incubated for 40 min at room temperature.

The chimeric structures for BAI1 were formed by mixing BAI1 conjugates with and without Jag1 conjugates in twenty times excess to each protruding site of the DNA nanostructure. BAI1 is a protein that interacts with integrin receptors and CD36[64]. In a previous study co-localization of integrins and Notch1 in neural progenitors indicates either direct interaction of the two proteins or implication of integrins in the Notch intracellular trafficking[65].

The DNA origami and conjugate mixtures were incubated in a PCR machine with a temperature ramp starting from 1 h at 37 °C followed by 14 h at 22 °C, and immediately after incubation, the nanopatterns were stored at 4 °C. The excess of conjugates that didn't react with the DNA origami were removed by using an FPLC system with a size

exclusion purification column (Superose 6 Increase 10/300GL, Cytiva). After purification, fractions of the peak corresponding to chimeric nanopatterns were collected and concentrated in 30-kDa MWCO 2-ml Amicon centrifugal filters (Millipore). The final sample concentration was estimated with agarose gel electrophoresis by loading the sample of known concentration before purification and the sample after purification of unknown concentration. By comparing the intensity of the bands we calculate the final concentration of each sample.

## Reporting summary

Further information on research design is available in the Nature Portfolio Reporting Summary linked to this article.

## Data availability

The detailed DNA origami design schematics including sequences has been deposited at nanobase.org under accession codes 192 and 231. The sequencing data generated in this study have been deposited at ArrayExpress under accession code E-MTAB-12439. Other data generated in this study are provided in the Source Data file. Source data are provided with this paper.

## Code availability

Computational code for RNA-seq analyzes and DNA-PAINT image analysis has been deposited at GitHub [https://zenodo.org/records/10178184].

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

## Acknowledgements

The authors would like to acknowledge support from the NIH grant number R35GM133482 for V.C.L., the Knut and Alice Wallenberg Foundation (Grants KAW 2017.0114 for B.H. and A.I.T. and KAW 2017.0276 for B.H.), from the European Research Council ERC for B.H. (Acronym: Cell Track GA No. 724872) and A.I.T (Acronym: Mech-Comm GA No. 617711), and from the Swedish Research Council for B.H. (grant no. 2019-01474) and from the Göran Gustafsson Foundation for B.H. And from the Academy of Finland for B.S. (grant no. 341908). lt-NES samples were obtained from, and initial culture protocols was made possible with the help of Anna Falk's team and the iPS Core facility at Karolinska Institutet. Part of this work was performed at the Karolinska Institutet/SciLifeLab Protein Science Core Facility (PSF). Part of this work was performed at the Karolinska Institutet Biomedicum Imaging Core (BIC). EM data was collected at the Karolinska Institutet 3D-EM facility.

## Author contributions

Conceptualization I.S., A.I.T., and B.H.; Methodology I.S. and B.H.; Investigation I.S., F.F., I.R.L., B.S., and Y.W.; Formal Analysis I.S., F.F., I.R.L., and A.L.; Resources V.C.L.; Funding acquisition B.H.; Supervision B.H., A.I.T., B.R.; Visualization I.S., I.R.L., Y.W., A.L., F.F., B.S., B.H.; Writing – original draft I.S., B.H.; Writing – review & editing I.S., F.F., V.C.L., B.R., A.I.T. and B.H.

## Funding

## Competing interests

The authors declare no competing interests.
