## [Peer Review File · Nature Communications]

Soluble and multivalent Jag1 DNA origami nanopatterns activate Notch without pulling forceREVIEWER COMMENTS

Reviewer #1 (Remarks to the Author):

The authors perform a comprehensive study of Notch signaling using DNA origami rods to organize the Jag1Fc ligand at varying copy numbers and spacing to investigate impact on downstream signaling. They use careful biochemical, super-resolution, and SPR characterization to validate their origami-protein constructs, including validation of precise nanoscale spatial patterning. Transcript expression analysis reveals interesting results such as highly sensitive behavior of the HEY1 gene triggered by a single ligand compared with other genes that show more monotonic behavior with increasing ligand number. They incorporate inhibitors or negative control experiments to test distinct activation mechanisms including electrostatics, non-specific surface attachment, a clathrin-mediated endocytosis inhibitor, and an ADAM inhibitor. Finally, they also incorporate chimeric structures with cholesterol replacing the ligand to test the impact of membrane-binding driving Notch activation.

Taken together, the authors present a comprehensive set of new data to offer insight into an alternative mechanism of Notch activation. This is a centrally important signaling cascade that is involved in a variety of basic biological and pathological processes, which has been under study for decades, and therefore should benefit from insight gained from these nanoscale ligand patterns and pertinent controls carried out by the authors.

My only minor suggestions for the authors to consider in revision are:

(1) Contemplate rewording line 273 from "Although the above results show a force independent activation of Notch..." to "Although the above results suggest a force-independent activation of Notch..."

(2) Related to this, consider slight editing of the Abstract sentence: "Here, we define conditions that show pulling force-independent Notch activation via soluble multivalent constructs" to something along the lines of, "Here, we define conditions that reveal/demonstrate pulling-force-independent Notch activation via soluble multivalent constructs" or similar.

and

(3) Consider incorporating in supplement a solution-based zeta-potential measurement of the charge of their bare DNA origami versus oligolysine-coated DNA origami, and perhaps also compare with a reference oligolysine-PEG-coated origami, to complement their gel-based analyses. This suggestion is made because the assumption of charge-neutralization by oligolysine appears quite central to the electrostatic repulsion hypothesis investigated in Figure 4, which may therefore benefit from additional solution-based quantitation of origami charge.

Reviewer #2 (Remarks to the Author):

The work by Smyrlaki¹ et al., investigates with a novel approach the mechanism of activation of Notch receptor. The mechanism of activation of Notch receptors is of importance both for biological understanding of this crucial signaling pathway during development, homeostasis and disease; and also for synthetic biology approaches aimed at engineering synthetic receptors based on the Notch paradigm.

The hypothesis put forward by the authors is that the exposure of the cleavage site for start of signaling could happen via an allosteric mechanism that does not require pulling force; the hypothesis is intriguing and could direct other researchers to investigate this aspect further, and could explain some strange results. At present, the main claims of the authors are only partially supported by the provided dataset, and as such I would require additional experiments to support the claims as follows:

1. The series of experiments in Fig. 5 are missing a control: generating a rod with one Jagged and with a few extra molecules that don't mediate interaction with the membrane (maybe GFP could be a choice?). I say this because everything you added to your 1x-Jag conjugated rod increases signaling (5b and 5e); so maybe it is a weight effect? This hypothesis would be supported also by the fact that there is substantial increase of signaling when you just moved Jag from the side to the center of the rod (as you notice in discussion, and is particularly obvious from the microscopy of Fig. 5d). Showing that adding an inert protein to the rod and that this does not increase signaling, would strengthen the argument that it is through increased time to the membrane, which right now is not strongly supported.
2. Experiment in Fig. 4c, with Pitstop; given it is a negative result, authors need to provide evidence that pitstop is indeed blocking endocytosis in their experiments.

A textual point that I would like to see addressed in a revision is the use of "force mediated" and "force-independent"; I understand where these terms come from, as in the field "force-mediated" refers to something where there is pulling-based exposure of the cleavage site; I think though that it is highly misleading in the current version to have a title that talks about force-independent when, for a non-expert reader, it could make very little sense what the authors are referring to.

The evidence that there is no pulling force is limited to the experiment with pitstop which has issues (see point 2. above), and even if properly controlled, does not exclude other non-clathrin mediated membrane rearrangements that could support pulling on a long rod with local membrane curvature for example (an hypothesis that would be supported by the fact that moving Jag in the center of the rod increases the signaling and the staining). Recommend tuning down the language around force-dependency.

Specific points:

lines 50-53; the summary of current theories does not encounter my full support, unless it is taken from a previous review or research papers. There are many theories for activation; point (i) of the statement is in my always part of all the theories that I know about, and as such I do not see the 2 presented theories

as alternative;

line 140: what does "model" mean here?

line 147: statement needs a reference

lines 212-217: are the references pointed at here looking at the same cellular system? would be good for the reader to know; one of the reasons for confusion in the Notch field is that there is a tendency to generalize taking results from a specific cellular system to make general claims, which I don't think is always appropriate;

lines 256-258, I don't see why was this expected? Why would there be changes in endocytosis upon signaling?

lines 273: example of use of "force-independent", not fully supported by data

Line 323-24: another example of strong language not fully supported by dataset

Line 335-353: I like this section a lot!

Line 390: it is not clear to me why at this point

Line 418: could be helpful for reader to expand on why is that

REVIEWER COMMENTS

Reviewer #1 (Remarks to the Author):

The authors perform a comprehensive study of Notch signaling using DNA origami rods to organize the Jag1Fc ligand at varying copy numbers and spacing to investigate impact on downstream signaling. They use careful biochemical, super-resolution, and SPR characterization to validate their origami-protein constructs, including validation of precise nanoscale spatial patterning. Transcript expression analysis reveals interesting results such as highly sensitive behavior of the HEY1 gene triggered by a single ligand compared with other genes that show more monotonic behavior with increasing ligand number. They incorporate inhibitors or negative control experiments to test distinct activation mechanisms including electrostatics, non-specific surface attachment, a clathrin-mediated endocytosis inhibitor, and an ADAM inhibitor. Finally, they also incorporate chimeric structures with cholesterol replacing the ligand to test the impact of membrane-binding driving Notch activation.

Taken together, the authors present a comprehensive set of new data to offer insight into an alternative mechanism of Notch activation. This is a centrally important signaling cascade that is involved in a variety of basic biological and pathological processes, which has been under study for decades, and therefore should benefit from insight gained from these nanoscale ligand patterns and pertinent controls carried out by the authors.

We would like to thank the reviewer for his/her work in reviewing our manuscript, and we are happy that the reviewer clearly appreciated our work.

My only minor suggestions for the authors to consider in revision are:

(1) Contemplate rewording line 273 from "Although the above results show a force independent activation of Notch..." to "Although the above results suggest a force-independent activation of Notch..."

We have rephrased the sentence to "Although the above results suggests that activation of Notch might occur without a pulling force being applied by the ligand, the question remained whether the multivalency effect we observed is due to clustering of the receptor or if the effect is mainly due to avidity and increased time-of-binding of ligand-receptor pairs (Fig. 5a)."

(2) Related to this, consider slight editing of the Abstract sentence: "Here, we define conditions that show pulling force-independent Notch activation via soluble multivalent constructs" to something along the lines of, "Here, we define conditions that reveal/demonstrate pulling-force-independent Notch activation via soluble multivalent constructs" or similar.

We rephrase the sentence to "Here, we define conditions that reveal pulling-force-independent Notch activation via soluble multivalent constructs"

and

(3) Consider incorporating in supplement a solution-based zeta-potential measurement of the charge of their bare DNA origami versus oligolysine-coated DNA origami, and perhaps also compare with a reference oligolysine-PEG-coated origami, to complement their gel-based analyses. This suggestion is made because the assumption of charge-neutralization by oligolysine appears quite central to the electrostatic repulsion hypothesis investigated in Figure 4, which may therefore benefit from

additional solution-based quantitation of origami charge.

We have now performed zeta potential measurements of bare, non-coated, and oligolysine (K10) coated DNA nanoparticles at a ratio of 0.5:1 nitrogen to phosphorus groups (N:P) in Lysine:DNA, and confirmed that the charge of the particles has indeed changed significant towards neutral in K10 coated nanoparticles (new Supplementary figure 6).

Reviewer #2 (Remarks to the Author):

The work by Smyrlaki1 et al., investigates with a novel approach the mechanism of activation of Notch receptor. The mechanism of activation of Notch receptors is of importance both for biological understanding of this crucial signaling pathway during development, homeostasis and disease; and also for synthetic biology approaches aimed at engineering synthetic receptors based on the Notch paradigm.

The hypothesis put forward by the authors is that the exposure of the cleavage site for start of signaling could happen via an allosteric mechanism that does not require pulling force; the hypothesis is intriguing and could direct other researchers to investigate this aspect further, and could explain some strange results. At present, the main claims of the authors are only partially supported by the provided dataset, and as such I would require additional experiments to support the claims as follows:

We would like to thank the reviewer for his/her work, we appreciate the fair points raised and the appreciation shown for our work.

1. The series of experiments in Fig. 5 are missing a control: generating a rod with one Jagged and with a few extra molecules that don't mediate interaction with the membrane (maybe GFP could be a choice?). I say this because everything you added to your 1x-Jag conjugated rod increases signaling (5b and 5e); so maybe it is a weight effect? This hypothesis would be supported also by the fact that there is substantial increase of signaling when you just moved Jag from the side to the center of the rod (as you notice in discussion, and is particularly obvious from the microscopy of Fig. 5d). Showing that adding an inert protein to the rod and that this does not increase signaling, would strengthen the argument that it is through increased time to the membrane, which right now is not strongly supported.

We interpret this comment as questioning whether inertia effects could play a role in the responses observed. To this end, an appropriate control would be to increase the molecular weight of the entire assembly without changing the interaction with the membrane.

To be able to alter the weight and at the same time be sure we are not adding any non-specific interactions with the cell surface we have opted for an experiment that we think addresses the reviewer's point even more accurately than adding GFP. We have instead added extra DNA helices to the DNA nanostructures and the added molecular weight from these extra base pairs (about 600 bp) corresponds to the same molecular weight as 3x Jag1Fc (3x 125 kDa). Therefore, a 1xJag NP with this extra DNA is equivalent to the normal 4x JagNP in terms of molecular weight.

These new results are presented in new Supplementary figure 8 and discussed in the manuscript after we discuss the Bai1 chimera structures.

As can be seen from these results, the increased molecular weight of the assembly does not increase signaling and we hope that this control experiment is in line with what the reviewer had in mind.

2. Experiment in Fig. 4c, with Pitstop; given it is a negative result, authors need to provide evidence that pitstop is indeed blocking endocytosis in their experiments.

We have now performed the requested control experiments for blocking endocytosis. These results are added in new Supplementary fig. 7 and referenced in the main text when we introduce the inhibitor experiment. These control experiments show significant blocking of endocytosis in the experimental setup we use.

A textual point that I would like to see addressed in a revision is the use of “force mediated” and “force-independent”; I understand where these terms come from, as in the field “force-mediated” refers to something where there is pulling-based exposure of the cleavage site; I think though that it is highly misleading in the current version to have a title that talks about force-independent when, for a non-expert reader, it could make very little sense what the authors are referring to.

The evidence that there is no pulling force is limited to the experiment with pitstop which has issues (see point 2. above), and even if properly controlled, does not exclude other non-clathrin mediated membrane rearrangements that could support pulling on a long rod with local membrane curvature for example (an hypothesis that would be supported by the fact that moving Jag in the center of the rod increases the signaling and the staining). Recommend tuning down the language around force-dependency.

We have now addressed the issues raised concerning the Pitstop experiment, see above. We think that the indications for force-independency go beyond the experiment with pitstop. First, we have eliminated several potential sources of pulling forces in fig. 4. Secondly, as we discuss in the manuscript, we do not believe that there is a rationale for pulling forces induced by membrane curvature to increase with multivalency of the nanopatterns. (Of note, each sample is normalized to the protein content, not structures, giving actually less protein in the higher order nanopattern samples. The yield of Jag attachment to 1x is >95%, yield of full attachment of 4x structures is less, but we use one quarter of sample of 4x compared to 1x JNPs). Additionally, receptor activation increased upon adding Bai1 or cholesterol to the nanostructures, which does not support a force-dependet mechanism. All in all, we do think that the aggregate of experiments presented is consistent with suggesting that a mode of activation could exist that is not dependent on the classical Notch pulling force.

We agree with the reviewer that we should have been more precise in our wording, using “pulling force” instead of just force and that the language should be toned down a bit as well as in the title, which we have done throughout in the revised manuscript. We prefer to keep the title somewhat similar, but toned down, like we suggest in this revision and hope the reviewer can agree with this formulation.

We also agree that it is not possible to exclude all types of vesicle mediated uptake, nor to completely block endocytosis with Pitstop 2. We have consequently toned down the claims following the Pitstop experiments and looking at endocytosis genes, please see comment further below.

The case of the middle position instead of the end position is indeed interesting, but in our view it rather complicates the picture about what this mode of activation could actually be, because as the reviewer correctly points out it is one data point that fit less well with our time-of-residence

explanation (which we also acknowledge in the discussion). But like we argue above, we do not think it implies that a pulling force is more likely to occur, nor does it change the bulk of the data we do get from the multivalency. But we agree that it is an interesting case that we hope to follow up in a future project.

Specific points:

lines 50-53; the summary of current theories does not encounter my full support, unless it is taken from a previous review or research papers. There are many theories for activation; point (i) of the statement is in my always part of all the theories that I know about, and as such I do not see the 2 presented theories as alternative;

We have changed the wording, the theories are no longer presented in opposition to each other.

line 140: what does "model" mean here?

We meant the type of cells were a model of neuronal progenitors. We have changed the wording to reflect this.

line 147: statement needs a reference

We have added a reference to: U. Marklund et al., Development. (2010)

lines 212-217: are the references pointed at here looking at the same cellular system? would be good for the reader to know; one of the reasons for confusion in the Notch field is that there is a tendency to generalize taking results from a specific cellular system to make general claims, which I don't think is always appropriate;

We fully agree with the reviewer and we have changed the wording to "...we observed that genes that have previously been shown in other cellular systems..." to point out that it could be a generalization.

lines 256-258, I don't see why was this expected? Why would there be changes in endocytosis upon signaling?

We have changed the wording here, to point out that this might not be expected, and softened the wording on the conclusions from the endocytosis inhibition and RNA seq of these genes as follows:

"It can be interjected that these basal function genes are expected to be reasonably stably expressed. However, nanoparticle uptake has previously been shown to change transcription of endocytosis genes(37), but here we do not observe this. Additionally, endocytosis inhibitors like the one we use here, are not expected to block endocytosis completely (Suppl. Fig. 7). Nonetheless, the fact that we observed that Jag1 multivalency determined Notch activation levels independently of endocytosis modulation suggests that these responses are not attributed to internalization effects."

lines 273: example of use of "force-independent", not fully supported by data

We have softened the wording here as well as in several other places.

Line 323-24: another example of strong language not fully supported by dataset
Changed also here.

Line 335-353: I like this section a lot!

Much appreciated!

Line 390: it is not clear to me why at this point

The statement is: “ Due to the fact that we are using Jag1Fc and not monomeric Jag1 binding domains, we cannot exclude that dimerization of Notch receptors are important for activation.”

What we wanted to say is that maybe dimers of Jag1 is a minimal pre-requisite for the effects we see – That since we did not do the experiments starting from monomeric Jag1 structures, it is not possible to exclude this possibility completely. We do think it is unlikely that dimers are required given the bulk of the experiments, but this caveat seemed appropriate to add. We couldn't figure out a much better way to say this so we have left the manuscript unchanged here.

Line 418: could be helpful for reader to expand on why is that

We have added: “...because in this theory the NRR region acts as a switch that only reveals the S2 cleavage site if pulled upon.”

REVIEWERS' COMMENTS

Reviewer #1 (Remarks to the Author):

The authors have addressed my comments so I'm happy to endorse publication of their valuable work.

Reviewer #2 (Remarks to the Author):

I am very pleased with the level of attention the authors put into the production of a revised manuscript based on my comments, both in the addition of more experiments, and in the textual revision.

The manuscript represents valuable progress in the field and I hope it will be shared as soon as possible with the rest of the community. Again, congrats for a the good work!